# A Nurse-Led Integrated Chronic Care E-Enhanced Atrial Fibrillation (NICE-AF) Clinic in the Community: A Preliminary Evaluation

**DOI:** 10.3390/ijerph19084467

**Published:** 2022-04-07

**Authors:** Brigitte Fong Yeong Woo, Wilson Wai San Tam, Taiju Rangpa, Wei Fong Liau, Jennifer Nathania, Toon Wei Lim

**Affiliations:** 1Alice Lee Centre for Nursing Studies, Yong Loo Lin School of Medicine, National University of Singapore, Singapore 119077, Singapore; nurtwsw@nus.edu.sg; 2Bukit Batok—Medical, National University Polyclinics, National University Health System, Singapore 659164, Singapore; rangpa_taiju@nuhs.edu.sg; 3Bukit Batok—Nursing, National University Polyclinics, National University Health System, Singapore 659164, Singapore; wei_fong_liau@nuhs.edu.sg; 4Department of Medicine, Yong Loo Lin School of Medicine, National University of Singapore, Singapore 119077, Singapore; mdcjn@nus.edu.sg; 5Department of Cardiology, National University Heart Centre, National University Health System, Singapore 119228, Singapore; toon_wei_lim@nuhs.edu.sg

**Keywords:** atrial fibrillation, nurse-led, integrated care, advanced practice nurse

## Abstract

The current physician-centric model of care is not sustainable for the rising tide of atrial fibrillation. The integrated model of care has been recommended for managing atrial fibrillation. This study aims to provide a preliminary evaluation of the effectiveness of a Nurse-led Integrated Chronic care E-enhanced Atrial Fibrillation (NICE-AF) clinic in the community. The NICE-AF clinic was led by an advanced practice nurse (APN) who collaborated with a family physician. The clinic embodied integrated care and shifted from hospital-based, physician-centric care. Regular patient education, supplemented by a specially curated webpage, fast-tracked appointments for hospital-based specialised investigations, and teleconsultation with a hospital-based cardiologist were the highlights of the clinic. Forty-three participants were included in the six-month preliminary evaluation. No significant differences were observed in cardiovascular hospitalisations (*p*-value = 0.102) and stroke incidence (*p*-value = 1.00) after attending the NICE-AF clinic. However, significant improvements were noted for AF-specific QoL (*p* = 0.001), AF knowledge (*p* < 0.001), medication adherence (*p* = 0.008), patient satisfaction (*p* = 0.020), and depression (*p* = 0004). The preliminary evaluation of the NICE-AF clinic demonstrated the clinical utility of this new model of integrated care in providing safe and effective community-based AF care. Although a full evaluation is pending, the preliminary results highlighted its promising potential to be expanded into a permanent, larger-scale service.

## 1. Introduction

Atrial fibrillation (AF) represents the most common sustained cardiac arrhythmia, whose spectrum ranges from paroxysmal, self-terminating episodes of electrical disturbance to a chronic condition with clinical sequelae. There is a 20 to 25% lifetime risk of developing AF among adults, and this reported value is similar between Caucasian and Chinese populations worldwide [1,2]. Nevertheless, this is likely an underestimation, since many living with AF may be asymptomatic and thus remain undetected [3]. 

Long-term outcomes of AF are clinically significant and manifold. Among AF patients, the elevated risk of heart failure and stroke has been found to translate into increased mortality [4]. Additionally, the quality of life (QoL) is impaired among AF patients, even when other cardiovascular conditions are absent [5]. Neuropsychiatric complications have also been reported, including vascular dementia and cognitive decline, as well as depressed mood and depression [6]. Collectively, such sequelae of AF exert a substantial burden on healthcare systems worldwide.

Accordingly, international [7] and regional [1] consensus guidelines have been published to standardise the management of AF. However, physicians’ compliance to such recommended guidelines appears to have been poor [8,9]. Poor adherence to consensus guidelines may have arisen from the lack of understanding of AF management or the presence of barriers to optimise clinical practice [10].

The prevention of stroke in the management of AF is complex since it requires frequent monitoring of the patient’s anticoagulation status and consequently the titration of the oral anticoagulation (OAC) dosage. In this context, clinicians have been observed to lack knowledge on AF and, hence, to demonstrate less confidence in making accurate clinical judgements [10]. Nonetheless, given its chronicity, AF warrants lifelong treatment, to which patient’s adherence critically influences optimal AF management, thereby highlighting the need for effective patient education [11].

While physicians may have knowledge about AF and stroke risk assessment in AF patients, such knowledge has not been translated into practical measures for preventing stroke. Physicians, especially non-cardiologist ones, tended not to initiate OAC therapy for AF patients when indicated [10].

Against the background of suboptimal AF management (both internationally and in Singapore), existing clinician-centric care models based in tertiary hospitals may be unable to address the rising prevalence of AF. This therefore underlines the need for more efficient clinical approaches. Accordingly, an integrated care approach has been recommended by the European Society of Cardiology for chronic AF patients [7]. Such a patient-centric integrated care model amalgamates crucial elements of patient care, engages patients in treatment plans and lifestyle modification, and encourages interdisciplinary collaboration [12].

Advanced practice nursing has a growing presence internationally [13]. The International Council of Nurses (ICN) (2020) defines an advanced practice nurse (APN) as a registered nurse (RN) who has developed expert knowledge, higher clinical competencies, and complex decision-making skills for extended practice [14]. The characteristics and scope of practice of a APN are crafted by the context in which the APN is credentialed to practice [14]. In Singapore, APNs are legally restricted to RNs who have completed the accredited Master of Nursing programme [15].

APN-led services across settings have been reported to be safe and exert positive impacts on patient-reported outcomes and clinical outcomes [16]. As alluded to previously, there is a rising tide of AF, and the clinical impacts of poorly managed AF can be debilitating. Stroke prevention is crucial to the management of AF, and the integrated model of care has been recommended for managing AF [7]. The integrated model for AF care has commonly been applied in the hospital-based setting [17,18]. It may be advantageous to evaluate whether APNs could lead integrated AF care in the community setting. This may reduce overcrowding of patients in the hospital by shifting the care for AF patients with lower acuity needs from the hospital-based outpatient specialist setting to the community setting.

### Objectives

This study aimed to provide a preliminary evaluation of the effectiveness of the Nurse-led Integrated Chronic care E-enhanced Atrial Fibrillation (NICE-AF) clinic in the community. The specific objectives were to evaluate its effectiveness in improving:AF patient-reported outcomes, i.e., AF-related QoL, AF knowledge, medication adherence, patient satisfaction and depression; andAF patients’ clinical outcomes, i.e., cardiovascular hospitalisation and stroke incidence.

The purpose of this preliminary evaluation was to assess if the NICE-AF clinic’s potential for development into a more permanent, larger-scale service.

## 2. Materials and Methods

### 2.1. Study Design

An uncontrolled before-and-after study design was adopted [19].

### 2.2. Setting

In this study, the NICE-AF clinic was based in Bukit Batok Polyclinic, which catered to the healthcare needs of residents in western Singapore. This facility was part of an integrated network of public outpatient healthcare centres (collectively known as the Polyclinics) responsible for delivering primary healthcare in Singapore. They provide subsidised medical care, post-hospital discharge medical follow-up care, immunisation, basic diagnostic services, pharmaceutical services, health education, and health screening. Apart from caring for acute illnesses (such as cough and cold) and acute injuries (such as minor fractures), the Polyclinics play a pivotal role in chronic disease management in the primary care setting through their multidisciplinary care teams. Such teams comprise physicians, nurses, and allied health professionals, including dieticians, medical social workers, pharmacists, psychologists, physiotherapists, and podiatrists [20]. APNs are also present in the Polyclinics, managing patients with chronic conditions and collaborating with physicians and allied health professionals in the multidisciplinary patient care.

### 2.3. Participants

Patients who met the following eligibility criteria were included as newly diagnosed AF, or chronic, stable AF (Table 1), as documented on an ECG, Holter, or cardiac event monitoring; at least 21 years old; able to provide informed consent; and able to converse in either English or Mandarin.

They were excluded if they were participating in another clinical study; or had pre-existing complex comorbidities requiring follow-up care by a hospital-based medical specialist; or had undergone cardiac surgery less than three months before recruitment; or were pregnant; or had underlying psychiatric disorders and were under treatment at the time of recruitment; or had cognitive decline or mental incapacitation.

### 2.4. Sampling

Consecutive sampling was adopted to recruit participants. In determining the sample size for this study, reference was made to a keynote Netherlands-based randomised controlled trial, which compared a nurse-led integrated chronic care approach with routine outpatient care for managing AF patients [18]. A hazard ratio of 0.66 was reported for the endpoint of cardiovascular hospitalisation, highlighting the superiority of the nurse-led care in reducing the risk of such hospitalisation by 34% [18]. Further reference to the literature revealed the hospitalisation rates among AF patients to be approximately 10 to 20% [17,18]. Hence, the correlation between the pre- and post-intervention cardiovascular hospitalisations was assumed to be 0.80 for this study. Accordingly, to detect a 34% reduction in cardiovascular hospitalisation at a power of 0.80 and statistical significance of 5%, 141 participants would be needed for this study. Assuming a 20% attrition, 170 participants had to be recruited. Nonetheless, recruitment was disrupted by the coronavirus disease 2019 (COVID-19) pandemic. This paper thus reports the preliminary evaluation of the 43 participants who completed the study (formal sample determination was not pursued).

### 2.5. Recruitment

Participants were recruited from the patient registry of Bukit Batok Polyclinic between September 2018 and June 2021. Physicians were encouraged to refer patients with an existing or new diagnosis of AF to the NICE-AF clinic. Upon such referral, they would be screened by the research team for eligibility for the study. However, between January and August 2020, recruitment was suspended in response to the COVID-19 pandemic, as healthcare institutions and the university deferred cross-institutional visits and non-critical research activities.

### 2.6. Intervention

The set-up of the NICE-AF clinic was guided by elements of the Chronic Care Model (CCM) (Figure 1) [21]. Its modality not only represented a shift from hospital-based physician-centric care, but also embodied integrated care management with elements such as technological tools for personalised, efficient, and accurate care [22]. The clinic served as a one-stop facility for patients to receive care for AF and common chronic conditions, including type II diabetes mellitus (DM), hypertension, and hyperlipidaemia.

In this study, the NICE-AF clinic was led by an APN. In the broader context of Singapore, APNs are master’s degree-prepared nurses with collaborative prescribing rights and limited referral and admission privileges [23,24]. Under the modality of the clinic (Figure 2), the APN, originally trained to deliver primary care, received additional training from a cardiologist to manage AF. Additionally, she collaborated with a family physician (FP) in delivering AF care. During NICE-AF clinic sessions, the APN used an electronic decision support flow sheet (customised for AF patients in Singapore) to guide and supplement her clinical decisions. A panel of experts in AF care had jointly formulated the electronic decision support flow sheet for managing chronic but stable AF in accordance to guidelines by the European Society of Cardiology [7] and the Asia Pacific Heart Rhythm Society [1].

During NICE-AF clinic consults, the APN delivered regular patient education, aided by a webpage (www.nice-af.wixsite.com/livingwithaf) on AF developed by the authors (B.W., L.T.W.). This online content was available in English and Simplified Chinese and covered AF management, anticoagulation, lifestyle activities, and diet modification. Additionally, the patients under this modality enjoyed fast-tracked appointments for hospital-based specialised investigations, including transthoracic echocardiogram (TTE), Holter studies, or treadmill ECGs. Lastly, consultations with a hospital-based cardiologist were available through telecommunication. The frequency of clinic visits is shown in Figure 3.

### 2.7. Outcome Measures and Data Collection

Patient-reported outcomes were measured with questionnaires validated in Singapore for both English and simplified Chinese. AF-specific QoL was measured through the Atrial Fibrillation Effect on QualiTy-of-life (AFEQT) questionnaire [25], AF knowledge through the Singapore Atrial Fibrillation Knowledge Scale (SGAFKS) [26], and patient satisfaction through the modified Patient Satisfaction Questionnaire (PSQ) [27]. Depression was screened, and its severity was quantified with the Patient Health Questionnaire (PHQ-9) [28]. For medication adherence, the participants were requested to self-report their frequency of forgetting their medications in the past seven days or past month.

The clinical outcomes, cardiovascular hospitalisation and incidence of stroke, were collected retrospectively (six months before study entry) and prospectively (six months after study entry) from the participant’s electronic medical records.

All questionnaires were administered face-to-face over two timepoints (T^0^ and T^1^): the first administration took place immediately before the first NICE-AF clinic consultation (T^0^), while the second took place six months after it (T^1^) (Figure 4). 

### 2.8. Data Analysis

Descriptive analysis was conducted to characterise the participants’ baseline demographic data and clinical data. The Wilcoxon signed-rank test was used to compare their clinical and patient-reported outcomes six months before recruitment into the NICE-AF clinic with those six months afterwards [29]. In this study, the level of significance was set at 5%. For those participants who dropped out or were lost to follow-ups, their data were omitted from analysis. For those who completed the questionnaires at timepoints T0 and T1, no missing data were noted. All data were analysed with the IBM SPSS (version 26.0, IBM corp., Armonk, NY, USA) [30].

### 2.9. Ethical Considerations

Approval was obtained from the National Healthcare Group Domain Specific Review Board (NHG DSRB ref: 2017/00682) for all the research activities involved in this study.

## 3. Results

### 3.1. Recruitment and Retention

Among the 559 patients screened (Figure 5), only 66 were eligible, of whom 11 declined to participate and the remaining 55 were recruited. Subsequently, 12 participants dropped out. Of these, six were voluntary dropouts; four were hospitalised in between NICE-AF clinic follow-ups and thereafter required specialist care; one required specialist care and was referred to a hospital-based cardiology clinic; and another requested a transfer to another Polyclinic closer to his home. Thus, by the time of this preliminary evaluation, 43 participants completed the six-month follow-up.

### 3.2. Characteristics of Participants

The socio-demographic profile of the 43 participants is outlined in Table 2. With a median age of 69 years (IQR 64–74 years), they comprised 29 males (67%) and 14 females (33%). The majority were Chinese (88%) and married (81%). A third lived with their spouse and children (37%), and another third lived with their children (37%). Approximately half of them had at least a secondary-school education (47%). Close to a third were in full-time employment, while half were retirees (51%).

The participants’ clinical profile is outlined in Table 3. The median duration of their AF diagnosis was 37.0 months (IQR 16.0–84.0). Their median CHA_2_DS_2_-VASc [congestive heart failure, hypertension, age ≥75 (doubled), DM, stroke (doubled), vascular disease, age 65–74, and sex (female)] score was 3.0 (IQR 2.0–4.0). Their median HAS-BLED [hypertension, abnormal renal/liver function (1 point each), stroke, bleeding history or predisposition, labile INR, elderly (>65 years), concomitant drugs/alcohol intake (1 point each)] score was 2.0 (IQR 2.0–3.0). Their comorbidities included hypertension (86%), type II DM (40%), ischaemic heart disease (16%), and congestive heart failure (2%).

### 3.3. Comparison of Patient-Reported and Clinical Outcomes between Timepoints T^0^ and T^1^

Five patient-reported outcomes were investigated (Table 4). For AF-specific QoL, a statistically significant improvement (*Z* = −3.29, *p* = 0.001) was demonstrated by its AFEQT scores: T_1(_ median 96.3, IQR 91.7–98.1) compared with T_0_ (median 93.5, IQR 88.0–96.3). Likewise, for AF knowledge, an improvement (*Z* = −4.70, *p*-value < 0.001) was demonstrated by its SGAFKS scores: T_1_ (median 6.80, IQR 5.60–8.60) compared with T_0_ (median 4.60, IQR 2.50–6.20). At the six-month follow-up, the participants reported significantly higher levels of QoL and AF knowledge than before receiving care at the NICE-AF clinic.

For medication adherence, a statistically significant improvement (*Z* = −2.67, *p*-value = 0.008) was demonstrated by the single-item questionnaire: T_0_ (median 1.00, IQR 1.00–2.00) compared with T_1_ (median 1.00, IQR 1.00–2.0). Likewise, for patient satisfaction, an improvement (*Z* = −2.33, *p*-value = 0.020) was demonstrated by its overall PSQ scores: T_0_ (median 4.00, IQR 4.00–4.00) compared with T_1_ (median 4.00, IQR 4.00–4.00). At the six-month follow-up, the participants reported significantly better medication adherence and patient satisfaction than before receiving care at the NICE-AF clinic.

Lastly, for depression, a statistically significant reduction (*Z* = −2.87, *p*-value = 0.004) was demonstrated by its PHQ-9 scores: T_0_ (median 0.00, IQR 0.00–1.00) compared with T_1_ (median 0.00, IQR 0.00–0.00). At the six-month follow-up, the participants reported significantly lower levels of depression than before receiving care at the NICE-AF clinic.

Two clinical outcomes were investigated (Table 5). For cardiovascular hospitalisations, there was no statistically significant difference (Z = −1.63, *p*-value = 0.102), although its range decreased from 3 to 0 between timepoints T_0_ and T_1_. Likewise, for the incidence of stroke, there was no statistically significant difference (Z = 0.00, *p*-value = 1.00). 

## 4. Discussion

This study preliminarily examined the effectiveness of an APN-led AF clinic adopting an integrated care approach towards community-based AF management. Emerging evidence from the results has demonstrated such a new modality could deliver safe and effective care to AF patients. Despite lacking improvements for cardiovascular hospitalisation and stroke incidence, the NICE-AF clinic has evidently contributed to substantial benefits for the numerous patient-reported outcomes.

In terms of patient-reported outcomes, the participants’ improved AF-specific QoL at the six-month follow-up at the NICE-AF clinic was reflected by the elevated overall median AFEQT score. Notwithstanding the already high baseline scores, the participants experienced further improvement in AF-specific QoL after receiving care at the NICE-AF clinic. It might be explained by the consistent delivery of patient education and counselling by the same team of healthcare professionals in the NICE-AF clinic about medications, treatment plans, and prevention of adverse events [31]. This had probably enabled them to be less anxious about their AF condition and cope better with it.

Additionally, the participants’ improved level of AF knowledge might have been intertwined with their improved AF-specific QoL [32]. Preliminary comparison of the SGAFFKS scores at the baseline and at the six-month follow-up has demonstrated a significant increase in their AF knowledge. This enhanced understanding of the AF diagnosis suggested the value of not only the patient education and counselling delivered during APN-led consultations at the NICE-AF clinic, but also the webpage created for them. However, this enhanced understanding might also have resulted from the Hawthorne effect [33]. Under regular reviews by the APN, the participants might have endeavoured to improve their health since they anticipated periodical assessments of their AF knowledge by the APN. This Hawthorne effect could exert a desirable impact on patients’ self-management and, hence, should be factored into integrated AF care.

For the participants’ medication adherence, the six-month preliminary evaluation likewise demonstrated a significant improvement at the six-month follow-up, despite the high baseline adherence. Besides the patient education and counselling at the NICE-AF clinic, some other socio-demographic factors might have promoted adherence. A large cross-sectional, Singapore-based study has found that older patients (mean 64 years, SD 10) and patients who were married or widowed were more likely to be adherent to their oral medications [34]. They tended to be retirees and have their families to remind them to take their medications. Both these two favourable socio-demographic features were exhibited by the NICE-AF clinic’s participants, most of whom were above the age of 64 (median 69 years) and either married or widowed (98%). Furthermore, the majority (92%) lived with their families, thereby increasing the likelihood of family support. Such support is integral to the care and medication adherence of patients with chronic illnesses, since the next of kin are instrumental in reminding them of their medication intake [34]. Furthermore, therapeutic approaches involving both the patients and their family members have been found to contribute to desirable and sustainable effects on their health behaviours [35]. Therefore, at the outset of this study, the participants were encouraged to bring their family members to the NICE-AF clinic, with whom the patient-education webpage was also shared. Nonetheless, it was noteworthy that, since the participants self-reported their adherence, recall and social desirability biases might have predisposed them to overestimating it [36].

For the overall patient satisfaction, a significant improvement was likewise reported. Two underlying reasons might have been at work. Firstly, the consistent delivery of care by the same healthcare providers [37] might have made the approach more personalised. Moreover, such continuity would establish a sense of patient–provider rapport [38]. Secondly, better care integration [39] might have boosted patient satisfaction. The participants of the NICE-AF could not only have all their chronic conditions reviewed alongside AF, but also refill their prescriptions. This consolidated care translated into the logistical convenience that they would not have to visit multiple clinicians.

For the depression scores, a reduction was reported in the six-month preliminary evaluation of the NICE-AF clinic, representing a significant improvement considering that the participants already had low depression scores at baseline. This finding paralleled that of the landmark randomised controlled trial examining the effectiveness of a nurse-led integrated model of AF management [32]. In the literature, depression has been shown to be a strong predictor of QoL in both patients with AF [40] and patients with other cardiovascular diseases [41]. In this preliminary evaluation, the lower depression scores corresponded with the improved AF-specific QoL, corroborating the findings from those two said studies. This further underscore the importance of addressing patients’ psychological well-being in AF management, with a focus on allaying depressive thoughts and thus on improving the QoL [41]. The APN-led integrated approach to AF care, where the APN plays a critical role in reassurance and counselling, emphasising on patient education and self-management, may likely have a positive impact on AF patients’ mental health as opposed to usual care where dedicated consultation time and competencies are invariably lacking to undertake these activities [32,42].

Besides the improvements observed in patient-reported outcomes, there were no adverse safety signals in the clinical outcomes as well. Instead, the six-month preliminary evaluation revealed a reduced frequency for cardiovascular hospitalisation (4 at baseline T_0_ versus 0 at T_1_) after the NICE-AF clinic. Nonetheless, the difference was not statistically significant (*p*-value 0.102). Similarly, the evaluation revealed no statistically significant difference between stroke incidence at six months before attending the clinic and that at six months afterwards. Such findings are echoed in the literature on outpatient nurse-led AF care [17,43,44,45,46]. Despite their use of more rigorous comparative methods in countries such as Australia [43], Canada [17], Denmark [44], and the Netherlands [45], the studies in the literature likewise reported no significant difference in cardiovascular hospitalisation between nurse- and physician-led AF care.

One reason for the inability to detect any improvements in cardiovascular hospitalisation and stroke incidences could be that the follow-up period of six months might not have been sufficient. A longer period of observation would have been more propitious if the purposes of the study were not to preliminarily evaluate the clinic. Another reason could be that the participating AF patients were generally stable and only needed low-acuity care. Their low baseline frequency of hospitalisations was reflected by a modest cardiovascular risk: a median CHA_2_DS_2_-VASc score of 3 (low to moderate risk for stroke) and a median HAS-BLED score of 2 (low to moderate risk for bleeding complications) [7]. Furthermore, this was corroborated by the fact that the participating AF patients had been recruited from a Polyclinic, which typically catered to community-dwelling patients not in need of complex care.

Given such modest disease severity, differences in the type of intervention might not exert any acute or immediate influences on clinical outcomes such as hospitalisation [45]. However, when AF patients develop more comorbidities, the role played by nurse-led integrated AF management in diagnostics and therapies might be accentuated over more extended periods [17,47,48]. In this context, at least 86% of the recruited participants were living with one other chronic illness, for whom the NICE-AF clinic might thus reduce preventable hospitalisations over time. 

On the whole, the positive impacts on the numerous patient-reported outcomes from the NICE-AF clinic have probably arisen from the consistent patient education and counselling by the APN. In general, patient education is inherent in nursing, even in basic preparatory nursing training. Nurses, therefore, have a professional role in patient education and are looked upon to incorporate teaching in most aspects of their practice [49]. Therefore, the presence of the APN, with her advanced expertise in clinical knowledge and nursing skills, is envisioned to provide high-quality patient education and counselling in the clinic [50]. Alongside the APN, another highlight was the team of healthcare providers, whose consistent presence enabled both more personalised care and patient–provider rapport [38]. The care integration also yielded the benefit that the participants could have all their chronic conditions reviewed alongside AF, with the consequent convenience of not needing to visit multiple clinicians.

### Limitations

Some limitations of this preliminary evaluation of the NICE-AF clinic are noteworthy. The study was limited by its design, in which the lack of a control group made it difficult to determine the causality between the intervention and the outcomes of interest [51]. The Hawthorne effect represented another limitation, given its confounding effect which could have led to an overestimation of the effectiveness of this modality for both clinical and patient-reported outcomes [52]. The participants’ foreknowledge that their progress would be monitored could have led to the effect.

For before-and-after studies, another limitation is the presence of temporal changes. Self-learning occurs naturally for most people. As patients live longer with their chronic conditions, they tend to become better at learning and coping with them over time; such improvement occurs in a manner independent of any interventions [53]. Under a before-and-after study design as in this study, it would be challenging to isolate these temporal changes from intervention-induced changes (i.e., the NICE-AF clinic). Thus, the effectiveness of the NICE-AF clinic might not have been conclusively ascertained. Nonetheless, the preliminary evaluation in this study has been insightful in assessing the potential of expanding the clinic into a larger-scale service.

## 5. Conclusions

The NICE-AF clinic, led by an APN, embodied an integrated model of care that represented a shift from hospital-based physician-centric care. The clinic served as a one-stop facility for patients to receive care for AF and common chronic conditions. The preliminary evaluation of the clinic has demonstrated the clinical utility of this new model of integrated care in providing safe and effective community-based AF care. The NICE-AF clinic appeared to exert positive impacts on AF patients’ AF-specific QoL, AF knowledge, medication adherence, patient satisfaction, and depression scores. It also appears to be safe in terms of clinical events. Although a full evaluation is pending, these preliminary results encouragingly suggest its promising potential for development into a more permanent, larger-scale service.

## Figures and Tables

**Figure 1 ijerph-19-04467-f001:**
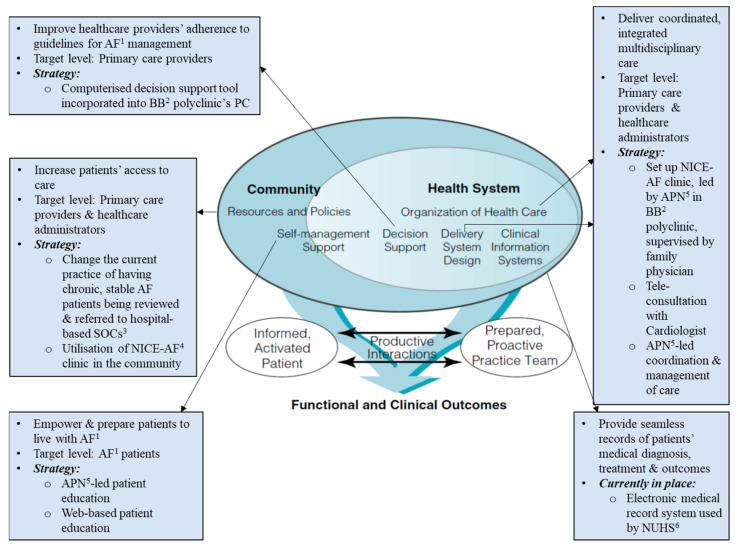
Elements of chronic care model guiding the design of the NICE-AF clinic. ^1^ Atrial fibrillation; ^2^ Bukit Batok; ^3^ specialist outpatient clinics; ^4^ Nurse-led Integrated Chronic care E-enhanced atrial fibrillation; ^5^ advanced practice nurse; ^6^ National University Health System.

**Figure 2 ijerph-19-04467-f002:**
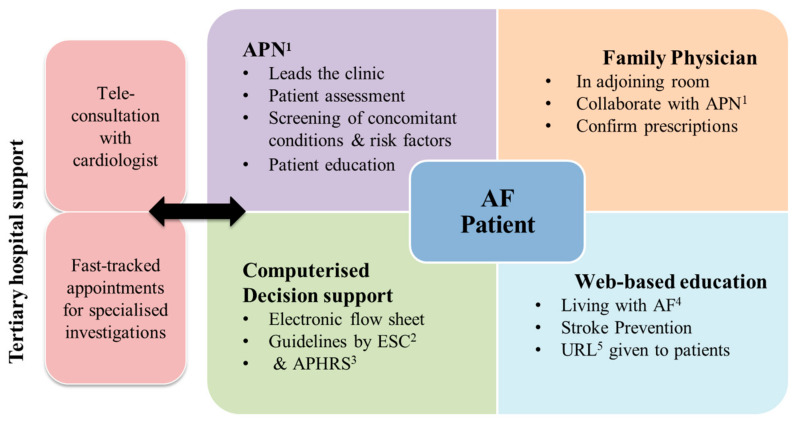
Outline of NICE-AF clinic. ^1^ Advanced practice nurse; ^2^ European Society of Cardiology; ^3^ Asia Pacific Heart Rhythm Society; ^4^ Atrial fibrillation; ^5^ uniform resource locator.

**Figure 3 ijerph-19-04467-f003:**
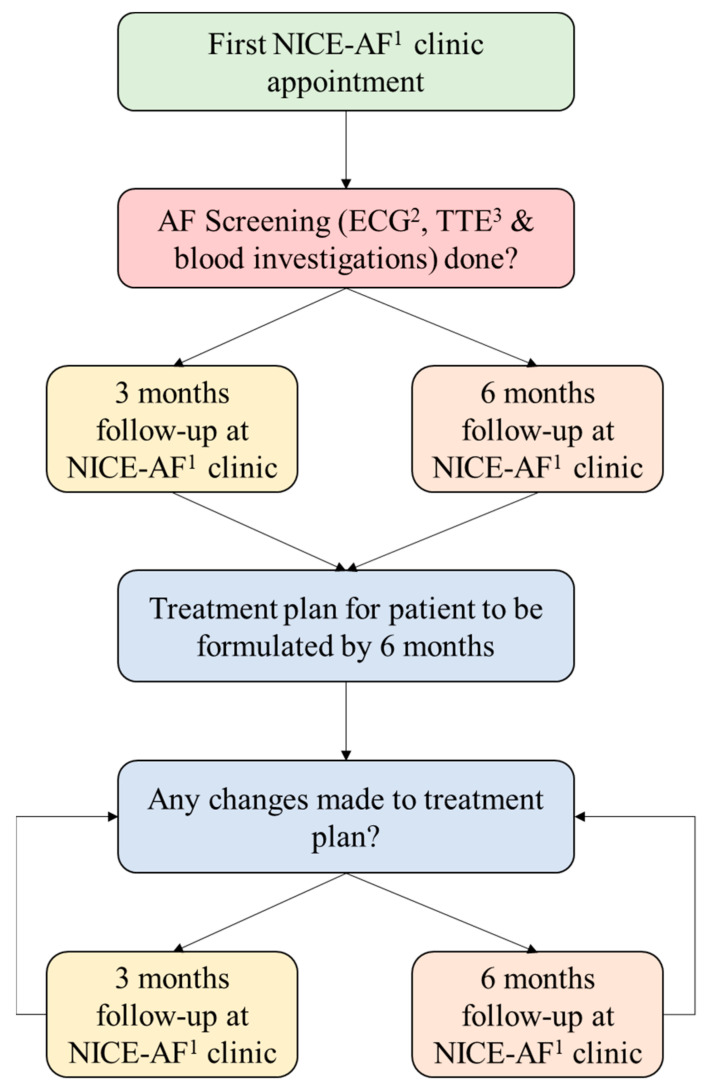
Frequency of clinic visits. ^1^ Nurse-led Integrated Chronic care E-enhanced Atrial Fibrillation clinic; ^2^ electrocardiogram; ^3^ transthoracic echocardiogram.

**Figure 4 ijerph-19-04467-f004:**
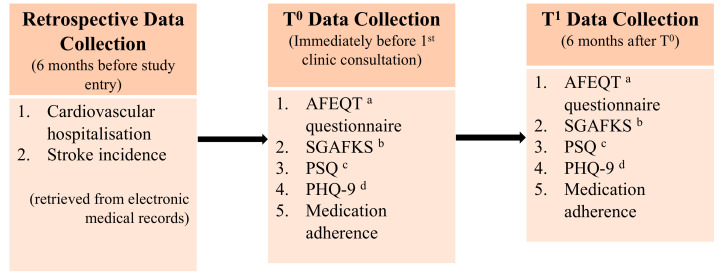
Outcome measures and data collection. ^a^ Atrial Fibrillation Effect on QualiTy-of-life; ^b^ Singapore Atrial Fibrillation Knowledge Scale; ^c^ Patient Satisfaction Questionnaire; ^d^ Patient Health Questionnaire.

**Figure 5 ijerph-19-04467-f005:**
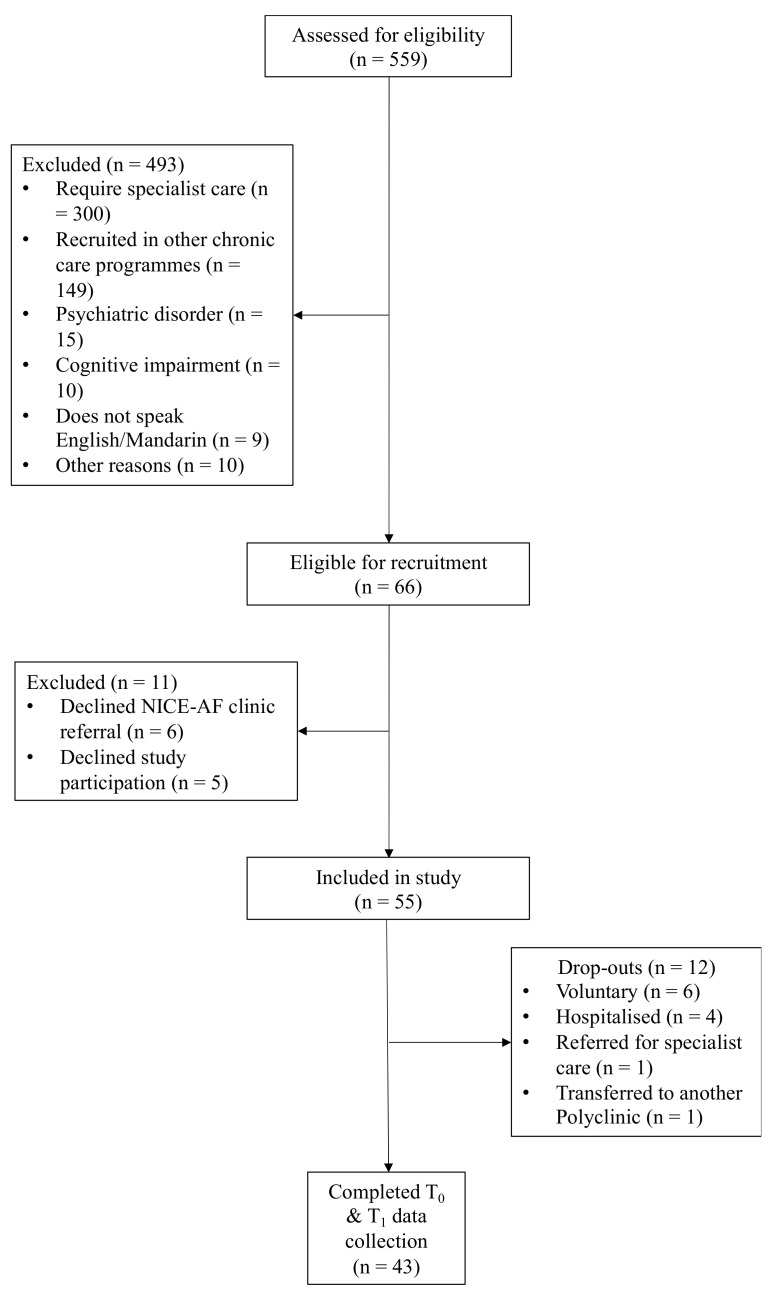
Recruitment process.

**Table 1 ijerph-19-04467-t001:** Operating definitions.

	Definitions
Stable AF	Defined as stable if the patient had: (a) not been hospitalised in the previous three months for cardiovascular diseases (including acute coronary syndromes, heart failure, arrhythmia, stroke, or systemic embolism); and (b) no change in anti-arrhythmia medications or anti-thrombotic therapy in the last three months.
Complex comorbidity	Includes: (a) uncontrolled hypertension that is being treated by an internist; (b) unstable heart failure defined as NYHA IV or heart failure necessitating hospital admission less than three months before recruitment; and (c) untreated hyperthyroidism or more than three months of euthyroidism.

**Table 2 ijerph-19-04467-t002:** Participant sociodemographic characteristics.

Sociodemographic Variables	AF Patients Included for 6-Month Comparison (*n* = 43)
Age (years), median (IQR)	69 (64–74)
Gender, *n* (%)	
Male	29 (67)
Female	14 (33)
Ethnicity, *n* (%)	
Chinese	38 (88)
Malay	5 (12)
Marital Status, *n* (%)	
Married	35 (81)
Widowed	7 (16)
Single	1 (2)
Living arrangement, *n* (%)	
Live with spouse and children	16 (37)
Live with spouse	16 (37)
Live with children	8 (18)
Live alone	3 (7)
Require caregiver, *n* (%)	0 (0)
Level of education, *n* (%)	
Primary school	14 (33)
Secondary school	20 (47)
Junior College/Pre-University	4 (9)
University	3 (7)
Pre-primary/no education	2 (5)
Employment status, *n* (%)	
Retired	22 (51)
Full-time	13 (30)
Part-time	6 (14)
Unemployed	2 (5)

**Table 3 ijerph-19-04467-t003:** Participant clinical characteristics.

Clinical Variables	AF Patients Included for 6-Month Comparison (*n* = 43)
Duration of AF (months), median (IQR)	37.0 (16.0–84.0)
CHA_2_DS_2_-VASc score, median (IQR)	3.0 (2.0–4.0)
HAS-BLED score, median (IQR)	2.0 (2.0–3.0)
Hypertension, *n* (%)	37 (86)
Type II DM, *n* (%)	17 (40)
Ischaemic heart disease, *n* (%)	7 (16)
Congestive heart failure, *n* (%)	1 (2)

**Table 4 ijerph-19-04467-t004:** Patient-reported outcomes at baseline and 6-month timepoints.

Outcome Variables	Baseline (T_0_) (n = 43)		6-Month (T_1_) (n = 43)			
	Median (IQR)	Mean ± SD	Median (IQR)	Mean ± SD	*Z* ^a^	*p*-Value
AF-specific QoL (AFEQT overall score)	93.5 (88.0–96.3)	90.0 ± 11.5	96.3 (91.7–98.1)	94.7 ± 5.4	−3.29	0.001 *
AF knowledge (SGAFKS score)	4.60 (2.50–6.20)	4.54 ± 2.5	6.80 (5.60–8.60)	6.91 ± 1.8	−4.70	<0.001 *
Medication adherence	1.00 (1.00–2.00)	1.49 ± 0.506	1.00 (1.00–2.00)	1.26 ± 0.441	−2.67	0.008 *
Patient satisfaction (PSQ overall score)	4.00 (4.00–4.00)	3.91 ± 0.366	4.00 (4.00–4.00)	4.07 ± 0.258	−2.33	0.020 *
Depression (PHQ-9)	0.00 (0.00–1.00)	1.05 ± 1.59	0.00 (0.00–0.00)	0.530 ± 1.44	−2.87	0.004 *

^a^ Related-samples Wilcoxon signed-rank test statistic. * Statistically significant at 5% level of significance.

**Table 5 ijerph-19-04467-t005:** Clinical outcomes at baseline and 6-month timepoints.

	Baseline (T_0_)^a^ (n = 43)	6-Month (T_1_) (n = 43)		
Outcome Variables	Median, Range	Median, Range	*Z* ^b^	*p*-Value
Cardiovascular hospitalisation	0, 3	0, 0	−1.63	0.102
Stroke incidence	0, 0	0, 0	0.00	1.00

^a^ On the basis of retrospective data (6 months before first NICE-AF clinic appointment) from participants’ electronic medical record. ^b^ Related-samples Wilcoxon signed-rank test statistic.

## Data Availability

The data presented in this study are available on request from the corresponding author. The data are not publicly available as the study data belong to the National University Health System, and institutional data-sharing restrictions are present.

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
