# Peer review of "A Nurse-Led Integrated Chronic Care E-Enhanced Atrial Fibrillation (NICE-AF) Clinic in the Community: A Preliminary Evaluation"

_ijerph, 2022, doi:10.3390/ijerph19084467_

Round 1

Author Response

Reviewer’s comments

Response(s)

1. What is the OAC (in lines 58)? The acronym needs to be extended when

appearing for the first time.

Apologies for missing this out the first time. OAC stands for oral anticoagulation. This has since been described in the revised manuscript on line 58.

2. “Their median CHA2DS2-VASc [congestive 232 heart failure, hypertension, age ≥ 75 (doubled), DM, stroke (doubled), vascular disease, 233 age 65–74, and sex

(female)] score was 3.0 (IQR 2.0–4.0).” Please remember about the proper way of

editing of score name in scientific articles.

Thank you for this feedback, it has since been edited to “CHA2DS2-VASc” across the entire manuscript.

3. “QualiTy-of-Life” (at line195), AFQET (at line 244 and in table 4) Is it a typo?

‘QualiTy-of-life’ should have been used instead to reply the ‘Q’ and the ‘T’ of the ‘AFEQT”. Thank you for pointing out. It has since been corrected.

4. “For the depression scores, a reduction was reported in the six-month preliminary

evaluation of the NICE-AF clinic, representing a significant improvement given the high baseline levels of depression. This finding paralleled that of the landmark

RCT examining the effectiveness of a nurse-led integrated model of AF management.” Could you explain this piece more? And, what is the RCT for the acronym?

This sentence has been reworded to bring better clarity. Please refer to line 337.

The acronym has been removed and spelt in full across the manuscript.

5. Similarly, the evaluation revealed no statistically significant 342 difference

between stroke incidence at six months before attending the clinic and that at 343

six months afterwards.

Whether the time of follow-up is enough to cause stroke? Could you explain more

description in the discussion?

Thank you for pointing this out. We believe that one of the key reasons that we could not detect any differences in stroke incidence before and after attending the clinic was that the participants were generally stable and only needed low acuity of care. We do also acknowledge that six months may not be a sufficient length of time to observe for stroke incidences. We have since expanded the discussion to include that. Please refer to line 411 of the revised manuscript.

6. In conclusion, ”The preliminary evaluation of the NICE-AF clinic has

demonstrated the clinical utility of this new model of integrated care in providing

safe and effective community-based 394 AF care.”

Could you explain more the comparison between the new model NICE-AF and

usual care? And cause good outcome.

Thank you for the feedback. We have since expanded the Conclusion section. Please refer to Line 420. We have decided to not over-emphasise the comparison between the NICE-AF clinic and usual care because this study is an preliminary evaluation and also undertook an uncontrolled before-and-after study design. As such, it would not be entirely fair to write a statement to say that this clinic is superior to usual care. The purpose of this preliminary evaluation is to assess its potential for development into a more permanent and larger-scale service.

Reviewer 2 Report

It is an attempt on community care to reduce overcrowding of patients in hospitals. It is a meaningful research to verify whether the burden of medical care can also be reduced.

Thanks for the opportunity to peer review.

General

The purpose of the research is clear and consistent.

A few retouching are required for positions that support community nursing.

It is an effect of working other than APN.

One is that the author should explain why it was a preliminary assessment.

Methods

1)The method is described specifically. APN education is also easy to understand.

But I couldn't find ethical considerations. How and by whom did you get your consent?

2)You checked the electronic records 6 months before T0. Six months ago, it is understandable that there is a figure about the relationship between T0 and T1.

I had to draw my own.

Results

1)Are the numbers in Figure 4 correct? Please check.

559-495=64?

2)N=55 is the point of T0. T1 decreased by 12 people to n=43. T0 in the last square confuses understanding.

Consideration

1) Is there a possibility that the number of eligible people was excluded by narrowing the number of eligible people to 1/10?

Are there very few people who have the potential to care for the community?

2) Emphasizes the effectiveness of counseling by APN. L371 also states that it is also an effect for other positions. Consideration should be made on the comparison of the effects of counseling between APN and non-APN.

Author Response

Reviewer comments

Response(s)

It is an attempt on community care to reduce overcrowding of patients in hospitals. It is a meaningful research to verify whether the burden of medical care can also be reduced.

Thanks for the opportunity to peer review.

Thank you for the positive feedback

General

The purpose of the research is clear and consistent.

A few retouching are required for positions that support community nursing.

It is an effect of working other than APN.

One is that the author should explain why it was a preliminary assessment.

We have conducted a preliminary evaluation of the NICE-AF clinic to assess if the clinic would have a positive impact on outcomes before scaling up the clinic’s services. This information has been included in the revised manuscript (page 3, line 100-102).

Methods

1)The method is described specifically. APN education is also easy to understand.

But I couldn't find ethical considerations. How and by whom did you get your consent?

Apologies for missing out the “ethics” section. It is now included on page 7 of the revised manuscript.

2)You checked the electronic records 6 months before T0. Six months ago, it is understandable that there is a figure about the relationship between T0 and T1.

I had to draw my own.

Thank you for the feedback. We have since heeded your advice and drew a new figure (i.e. Figure 4) to illustrate the data collection process. Please refer to page 7 of the revised manuscript.

Results

1)Are the numbers in Figure 4 correct? Please check.

559-495=64?

There was a minor typographical mistake which has since been corrected. Please refer to Figure 5 (previously titled Figure 4) of the revised manuscript.

2)N=55 is the point of T0. T1 decreased by 12 people to n=43. T0 in the last square confuses understanding.

The last square and final number of participants included for analysis was 43. We only included the results for the participants who completed both T0 and T1. We hope this clarifies.

1) Is there a possibility that the number of eligible people was excluded by narrowing the number of eligible people to 1/10?

Are there very few people who have the potential to care for the community?

For this preliminary evaluation, we had to exclude AF patients who were already enrolled in other chronic care programmes. However, considering the results from this preliminary evaluation, there are plans to move this group of patients from the chronic care programmes to the NICE-AF.

2) Emphasizes the effectiveness of counseling by APN. L371 also states that it is also an effect for other positions. Consideration should be made on the comparison of the effects of counseling between APN and non-APN.

Thank you for your suggestion to make a comparison between the effects of counselling between APN and non-APN. However, we have decided to leave this comparison out as it would deviate from the main intent of the study and discussion. In addition to highlighting the ability of the APN to lead this service, we wanted to highlight the usefulness of having a consistent team of healthcare providers to render a holistic and integrated care experience.

Round 2

Reviewer 1 Report

The author replied to the questions and clarified my doubt. 

Reviewer 2 Report

The author fully responded to my opinion.